# The Virus-Induced Transcription Factor SHE1 Interacts with and Regulates Expression of the Inhibitor of Virus Replication (IVR) in *N* Gene Tobacco

**DOI:** 10.3390/v15010059

**Published:** 2022-12-24

**Authors:** Ju-Yeon Yoon, Eseul Baek, Mira Kim, Peter Palukaitis

**Affiliations:** 1Department of Horticulture Sciences, Seoul Women's University, Seoul 01797, Republic of Korea; 2Department of Plant Protection and Quarantine, Jeonbuk National University, Jeonju 54896, Republic of Korea; 3Department of Agricultural Convergence Technology, Jeonbuk National University, Jeonju 54896, Republic of Korea

**Keywords:** cucumber mosaic virus 1a protein, inhibitor of virus replication, SHE1, SHE1-IVR co-localization, SHE1-IVR interaction, signaling pathways, transcription factor

## Abstract

The transcription factor SHE1 was induced by tobacco mosaic virus (TMV) infection in tobacco cv. Samsun NN (SNN) and SHE1 inhibited TMV accumulation when expressed constitutively. To better understand the role of SHE1 in virus infection, transgenic SNN tobacco plants generated to over-express SHE1 (OEx-SHE1) or silence expression of SHE1 (si-SHE1) were infected with TMV. OEx-SHE1 affected the local lesion resistance response to TMV, whereas si-SHE1 did not. However, si-SHE1 allowed a slow systemic infection to occur in SNN tobacco. An inhibitor of virus replication (IVR) was known to reduce the accumulation of TMV in SNN tobacco. Analysis of SHE1 and IVR mRNA levels in OEx-SHE1 plants showed constitutive expression of both mRNAs, whereas both mRNAs were less expressed in si-SHE1 plants, even after TMV infection, indicating that SHE1 and IVR were associated with a common signaling pathway. SHE1 and IVR interacted with each other in four different assay systems. The yeast two-hybrid assay also delimited sequences required for the interaction of these two proteins to the SHE1 central 58-79% region and the IVR C-terminal 50% of the protein sequences. This suggests that SHE is a transcription factor involved in the induction of IVR and that IVR binds to SHE1 to regulate its own synthesis.

## 1. Introduction

Various plant effector molecules, which interact directly with viral components to inhibit infection, are induced during virus infection [1,2,3,4]. These include RNA-dependent RNA polymerases (RDRs) [1,5], pathogenesis-related proteins [2,6], the inhibitor of virus replication (IVR) [1,7] and Argonautes plus other RNases [3,8]. Most of these factors are induced by phytohormones [4,9,10,11]. RDR1 is induced by several phytohormones involved in defense responses, including salicylic acid (SA), jasmonic acid (JA), ethylene (ET) and abscisic acid [12]. Previously, it was shown that infection of tobacco (*Nicotiana tabacum* cv. Samsun NN; SNN) by potato virus Y (PVY) stimulated RDR1, as well as several other defense-related genes, namely, alternative oxidase 1 (AOX1), RDR6, IVR and the transcription factor (TF) SHE1 (previously called ERF5) [13]. On the other hand, silencing the expression of RDR1 in transgenic tobacco plants either reduced the level of expression or prevented the expression of these defense-related genes, suggesting that RDR1 had the ability to regulate the expression of such genes [13]. AOX1 was previously known to be upregulated in tobacco by SA, although in a pathway independent of RDR1 induction by SA [14], whereas SHE1 was shown not to be regulated by SA, JA or ET [15]; SA was considered to be unnecessary for induction of IVR (unpublished work given in [1]). More recently, gene expression of *RDR1*, *SHE1* and *AOX1* was shown to be upregulated during infection of tobacco by cucumber mosaic virus (CMV), potato virus X (PVX), PVY and tobacco mosaic virus (TMV), although at different amplitudes and to different extents via the different viruses [16,17].

IVR and SHE1 are less well-known factors in the pathogen-defense signaling mechanism, although considerable research was conducted on the biology and biochemistry of IVR. IVR is a 21.6 kDa protein [18] that has been shown to be induced after transfection of (SNN) tobacco protoplasts with TMV. After IVR was purified from the culture medium of TMV-infected protoplasts and added to other NN tobacco protoplasts transfected with TMV, it inhibited virus replication [19]. The same inhibitory effect was demonstrated with several viruses (TMV, CMV, PVX and PVY) when applied to either tobacco leaf disks or whole plants recently infected [20,21]. IVR recovered from the intercellular fluid of TMV-infected NN tobacco leaves showed the same inhibitory effects [22]. Constitutive transgenic expression of cloned IVR, under the control of a cauliflower mosaic virus 35S RNA promoter, showed similar levels of resistance to infection by TMV in Samsun tobacco plants normally susceptible to infection (SNN) [23], relative to IVR applied to TMV-infected plants. The in vivo-expressed IVR protein showed a different electrophoretic mobility (c. 23 kDa) to the bacterially expressed IVR protein, suggesting that IVR was post-translationally modified [18]. IVR did not exhibit single-stranded RNase activity [20]. 

The 27 kDa SHE1 protein was first identified as an AP2/ERF class of TF, after isolation as a new TF able to bind (weakly) to GCC box *cis*-elements [15]. TMV infection induced SHE1, and transgenic over-expression of this TF was able to reduce the infection by TMV in SNN tobacco, both locally and systemically (the latter at a non-restrictive temperature, i.e., above 28 °C), apparently through a novel signal-transduction mechanism [15]. Later, SHE1 and IVR were analyzed for changes in transcription when nontransformed and RDR1 silenced tobacco plants were infected by either TMV (locally) or PVY (systemically) [13]. More recently, we showed that the expression of *SHE1* was induced constitutively in transgenic tobacco plants expressing the CMV 1a protein, and that the CMV 1a protein interacted with SHE1, perhaps to prevent SHE1 from functioning in resistance to CMV [17]. To gain a better understanding of the role of SHE1 in plant defense against viruses, especially TMV, we examined the effect of silencing or over-expressing *SHE1* in transgenic tobacco, whether SHE1 was involved in the same unknown pathway as IVR and/or whether SHE1 and IVR interacted with each other.

## 2. Materials and Methods

### 2.1. Plants and Virus Materials

*N. tabacum* cv. SNN plants, either nontransformed or transgenic for expression of the CMV 1a protein [24], over-expressing the *SHE1* gene, or silenced for expression of the *SHE1* gene (described below), were used for virus inoculation experiments and protoplast preparation. *N. benthamiana* was used for bimolecular fluorescence complementation (BiFC) analyses and the Duolink Proximity Ligation Assay (PLA). The sources of the viruses, TMV-U1, Fny-CMV, potato virus X (PVX-Kr) or potato virus Y (PVY-O) were as described in [16].

### 2.2. Plant Culture and Virus Inoculation

Plant propagation and maintenance at 25 °C, as well as virus inoculation, were all carried out as described previously [17]. For experiments conducted at 33 °C, the plants were maintained in a growth chamber with a 16 h light/8 h dark cycle [25]. The leaves of tobacco SNN plants, at 4-weeks old, were dusted with Carborundum and then inoculated by rubbing with a buffered extract of virus diluted in 0.01 M phosphate buffer, pH 7.2. All plant experiments were conducted either three times or with multiple plants (or cells) assessed in a particular analysis.

### 2.3. Plasmid Construction and Plant Transformation

The entire cDNA fragment of *SHE1* was obtained as a polymerase chain reaction (PCR) product from plasmid p1aX1-2 using specific primers, SHE1-For (5’-GAGA*GGATCC*ATGTCAAGTAACTCAAGCCCAC-3’) and SHE1-Rev (5’-GAGA*GAGCTC*TCAGTCCCTT CGACACGAATGT-3’). The PCR product was cloned into the pROK2 vector double-digested with *Bam*H1 and *Sac*I. The inverted-repeat PCR fragment obtained from the *SHE1* gene was ligated into the RNAi vector pFGC5941 (see Appendix A) and subsequently digested with *Asc*I/*Swa*I or *Xba*I/*Bam*HI to generate two complementary dsRNA fragments to induce the small RNAs of SHE1 [26]. Plasmids expressing *SHE1* from the vector pROK2 and a hairpin *SHE1* construct from the vector pFGC5941 were introduced into *N. tabacum* SNN using *Agrobacterium tumefaciens* strain LBA4404, according to standard procedures. Transgenic tobacco plants were regenerated as described previously [24].

The tobacco SNN IVR cDNA clone (NC330; [18]) was generously provided by G. Loebenstein (Volcani Inst., ARO, Israel), now deceased. The full-length or partial cDNAs of *SHE1* and *IVR* were amplified by reverse-transcription-PCR (RT-PCR), for use in the yeast two-hybrid (Y2H assay) with *SHE1* and *IVR*, with the primer sequences listed in Appendix A (see also Appendix A) and constructed in pAS2-attR (Bait) and pACT2-attR (Prey) vector using the Gateway LR reaction (Invitrogen, Carlsbad, CA, USA) via pDONR207. 

### 2.4. RNA Extraction, RT-PCR, RT-qPCR and Northern Blot Hybridization Analyses 

RNA extraction and purification from nontransformed and transgenic tobacco plants, as well as RT-PCR analyses, were all carried out as described previously [17], using the primers listed in Appendix A. Semi-quantitative RT-PCR was performed at 25 to 40 cycles to detect gene expression of *SHE1* and *IVR* in transgenic and nontransformed tobacco plants, as previously described [13]. Densiometric analyses of RT-PCR products were performed using the program ImageJ (image processing and analysis in Java; htpps://imagej.nih.gov/ij/index/html (accessed on 20 December 2022)). Quantitative RT-PCR (RT-qPCR) was performed as described by Baek et al. [16], using tobacco eIF4E as a reference gene.

For Northern blot analysis, total RNA was separated on either a denaturing agarose gel containing 6% formaldehyde or by semi-denaturing 15% PAGE (in 7 M urea) for small interfering RNAs (siRNAs). After blotting to nitrocellulose membranes, the siRNAs were hybridized to digoxigenin-labeled RNA probe against SHE1. The SHE1 probe was obtained by linearization of plasmid p1aX1-2 with *Spe*I and transcription using T7 RNA polymerase (Thermo Fisher Scientific, Carlsbad, CA, USA). The membranes were exposed to X-ray films (Kodak Biomax Light Film, Sigma-Aldrich, St. Louis, MO, USA) in the darkroom.

### 2.5. Characterization of Transgenic Plants

The transformed tobacco plants were assessed for either constitutive expression of *SHE1* (OEx-SHE1 lines) or *SHE1* siRNA production for the silencing-induced plants (si-SHE1 lines) by Northern blot hybridization (Appendix A). T1-generation plants, of lines SNN-OEx-SHE1#5 and SNN-si-SHE1#4, were chosen for the experiments conducted in this work.

### 2.6. Protein Extraction, Co-Immunoprecipitation and Western Blot Analysis

To determine whether SHE1 and IVR interact in planta, mRFP-tagged SHE1 (pROK2-SHE1-mRFP) and HA-tagged IVR (pGWB415-IVR-HA) were co-expressed in *N. benthamiana* by agroinfiltration and confirmed by co-immunoprecipitation. Proteins were extracted from leaves harvested 3 days after agroinfiltration, together with empty vector infiltrated control leaves. Total protein was extracted from 1 g of frozen leaf powder in 1 mL of extraction buffer (25 mM Tris-HCl, pH 7.5, 150 mM NaCl, 1 mM EDTA, 10% glycerol, plus one tablet of protease inhibitor mix (Roche Diagnostics, Laval, QC, Canada)) and then incubated with Protein G Magnetic Beads (Pierce Biotechnology, Rockford, IL, USA) plus anti-HA monoclonal antibody (Santa Cruz Biotechnology, CA, USA) at room temperature for 1 h, with mixing. The beads, coupled with the antibody and the target protein, were washed with TBS-T (Tris-buffered saline, 0.05% Tween 20) while gently mixing and eluting in distilled water, according to the manufacturer’s instructions. Immunoprecipitation followed by Western blot analysis was performed to detect the target protein by the commonly used method [17]. The eluted proteins were separated on a 12% SDS-polyacrylamide gel, by electrophoresis at 120 V, and then transferred to nitrocellulose membranes (Bio-Rad, Hercules, CA, USA), using the Mini-PROTEAN Tetra electrophoresis system (Bio-Rad) at 300 mA for 1 h. The membranes were blocked with PBS-T (phosphate-buffered saline, 0.1% Tween 20) containing 5% nonfat milk and incubated with the antisera against either IVR or SHE1 (both diluted 1:1000 in PBS-T), obtained from the Plant Virus GenBank, Seoul, Korea. The membranes were probed by anti-rabbit IgG secondary antibody conjugated with alkaline phosphatase (Promega, Madison, WI, USA) and developed in Western Blue® Stabilized substrate for alkaline phosphatase (Promega) at room temperature for 10 min.

### 2.7. Yeast Two-Hybrid and X-Gal Assays

Y2H assays for interaction between proteins expressed by plasmid constructs containing the full-length or partial C-terminally deleted *SHE1* genes (in the pAS2 vector) and full-length or partial N-terminally deleted *IVR* genes (in the pACT2 vector) were performed as follows. Constructs of bait and prey were co-transformed into the yeast strain Y2HGOLD, from the Matchmaker® Yeast Two-Hybrid System (Clontech, 630489) (Clontech, Mountain View, CA, USA), according to the manufacturer’s instructions. The transformants were selected on SD/-Leu, SD/-Trp, SD/-Leu/-Trp and SD/-Leu/-Trp/-His plates, essentially as described [17]. Protein–protein interactions were confirmed on SD/-Leu/-Trp/-His plates containing X-α-gal.

### 2.8. BiFC and Duolink In Situ PLA

The modified and enhanced yellow fluorescent protein (YFP) system [27,28] was used for BiFC, as described previously [17]. The SHE1 construct, with the N-terminal half of YFP at the N-terminus of SHE1, was prepared as described previously [17], while the IVR construct, with the C-terminal half of YFP at the C-terminus of IVR was prepared by the same method, using the primers shown in Appendix A.

Duolink in situ PLA was performed using the Duolink PLA kit (Abnova, Taipei, Taiwan) as described previously [29,30], in this case to detect the interactions between SHE1 and IVR. Protoplasts were isolated from leaves of 6-week-old *N. benthamiana* plants agroinfiltrated with plasmids harboring SHE1-HA and IVR-FLAG. The cDNAs of SHE1-HA and IVR-FLAG were cloned into pGWB415 and pGWB412 using the Gateway system [31]. The protoplasts were fixed and incubated with anti-HA mAb according to the manufacturer’s instructions. PLA detection using mouse anti-FLAG mAb was conducted and the fluorescence images were observed with a laser scanning confocal microscope (Nikon C2 model, Nikon Instruments, Tokyo, Japan). Texas red (TX2-red) was visualized with excitation laser at 561 nm, and blue signals (UV) were visualized with excitation laser at 405 nm. 

## 3. Results

### 3.1. Expression of SHE1 in Various Transgenic Lines: Effect of Virus Infection

Transgenic tobacco SNN plants generated to either over-express (OEx-SHE1) or silence the expression of *SHE1* (si-SHE1), as well as nontransformed tobacco SNN and transgenic tobacco expressing the CMV 1a protein (SNN-1a), were inoculated with TMV, PVX, CMV or PVY, and three days post inoculation (dpi), these plants were assessed for the expression of *SHE1* by RT-PCR (Figure 1). Previously, we showed that transgenic expression of the CMV 1a protein led to constitutive expression of SHE1, but at a very low level [17], not detectable here in either mock-inoculated SNN-1a plants or such plants infected with CMV or PVY at 3 dpi, but detectable in SNN-1a plants infected with TMV or (to a lesser extent) PVX (Figure 1). In fact, except for infection with PVY, the level of SHE1 mRNA was similar in virus-infected nontransformed plants as in virus-infected SNN-1a plants (Figure 1). This experiment also verified that the transgenic plants generated to either silence or over-express the *SHE1* gene did, in fact, show those expected accumulation phenotypes, and that infection by TMV and PVY further enhanced the expression of *SHE1* in the OEx-SHE1 line (Figure 1).

As seen in Figure 1, the RT-PCR products for OEx-SHE1 samples showed double bands. This is due to the transgene being derived from tobacco cv. Xanthi-nc, whereas the transformed plant was tobacco cv. SNN. As was pointed out by Yoon and Palukaitis [17], the SHE1 ORFs in SNN and Xanthi-nc are of different lengths, with the SNN SHE1 ORF ending 24 nt after the Xanthi-nc SHE1 ORF (see Appendix A of [17]). 

### 3.2. Effect of Silencing or Over-Expressing SHE1 on Movement of TMV

Since TMV showed the greatest effect of the four viruses on activating SHE1 transcription in any of the inoculated plants, regardless of the transgene status (Figure 1), we examined what effect TMV infection had on the local lesions produced in these NN tobacco lines that also expressed resistance against infection by TMV via a hypersensitive response. This was performed on leaves of tobacco SNN, nontransformed or transgenic for OEx-SHE1 or si-SHE1, after inoculation with wildtype (WT) TMV (Figure 2A). Silencing of SHE1 had no effect on either the number (Figure 2B) or the size (Figure 2C) of the local lesions, but over-expressing SHE1 led to smaller (Figure 2B) and fewer (Figure 2C) lesions. There was also a qualitative change in the appearance of the lesion in the OEx-SHE1 plants (Figure 2A, middle leaf vs. outer leaves). The effect of over-expression of SHE1 on lesion diameter was noted before [15]. Previously, it was shown that the expression of the CMV 1a protein did not increase the size of local lesions induced by WT TMV infection [25], and thus, these were not measured again here. In tobacco SNN inoculated with TMV-GFP, no visible lesions appeared (Appendix A), although this is cultivar dependent, since smaller lesions appeared on tobacco cv. SR1:NN and microscopic fluorescent lesions appeared in tobacco cv. Samsun NN [25]. In SNN-1a plants, TMV-GFP produced visible local lesions, smaller than produced by WT TMV (Appendix A); these were delayed in appearance by one day [25].

In tobacco plants grown above 28 °C, the resistance induced by the *N* gene was not operational against WT TMV [32,33] but still functioned in blocking systemic infection by TMV-GFP, although this resistance could be overcome in SNN-1a plants [25]. Therefore, here, we examined whether the resistance still present at higher temperatures could affect the systemic movement of TMV-GFP in OEx-SHE1 and si-SHE1 transgenic tobacco plants (Figure 3). At 2 weeks pi (wpi), the control plants, SNN and SNN-1a showed that the local spread of TMV-GFP was enhanced in SNN-1a plants vs. SNN plants, and that there was also systemic spread of TMV-GFP into upper leaves of SNN-1a plants but not in SNN plants (Figure 3, upper two panels), confirming previous results [25]. By contrast, the OEx-SHE1 plants showed a weaker accumulation of TMV-GFP in the inoculated leaves and no detectable systemic accumulation (Figure 3, lower left panel), whereas the infected si-SHE1 plants were similar to the infected control SNN plants, with regard to local accumulation and no systemic accumulation of TMV-GFP (Figure 3, lower-right panel vs. upper-left panel). Thus, the reduction in accumulation of SHE1, either in the control SNN plants at 33 °C [15] or in the *SHE1* silenced tobacco plants (Figure 3), did not overcome the barrier to systemic infection by TMV-GFP, as did expression of the CMV 1a protein (Figure 3; and [25]), indicating that the resistance to systemic movement of TMV-GFP (but not WT TMV) is not a function of SHE1 expression.

Silencing the *SHE1* gene expression also had an effect on systemic infection by WT TMV at ambient temperature, with a few lesions and other necrosis leading to leaf distortion occurring in upper leaves at later times post infection (37 dpi) (Figure 4A, right panel vs. left panel). Although asymptomatic at earlier times, the presence of TMV RNA and TMV capsid protein was confirmed by RT-PCR and Western blot analysis, respectively, in upper leaves, even at 17 dpi (Figure 4B, right panels). The data in Figure 3 and Figure 4 suggest that the absence of SHE1 has less of an effect on the local movement of either WT TMV or TMV-GFP, but rather allowed a slow systemic infection by WT TMV to occur.

### 3.3. Effect of Silencing or Over-Expressing SHE1 on IVR mRNA Accumulation

To determine whether SHE1 might be a TF in the pathway leading to or affecting IVR mRNA accumulation, RNAs extracted from the transgenic lines silencing or over-expressing SHE1, and either non-inoculated or infected with TMV, were examined by both RT-qPCR (Figure 5A,B; Appendix A) and semi-quantitative RT-PCR (Figure 5C). The results of infection by TMV in SNN tobacco plants showed an increase in accumulation of both SHE1 and IVR mRNAs (Figure 5A,B), as expected [7,13,15], with no detectable accumulation of SHE1 mRNA but a very low level of IVR mRNA detectable in the mock-inoculated plants (Figure 5C). In OEx-SHE1 tobacco plants, both SHE1 and IVR mRNAs were expressed constitutively and SHE1 mRNA, and to a lesser extent, IVR mRNA, were expressed at a higher level after infection with TMV (Figure 5C). By contrast, in si-SHE1 tobacco plants, accumulation of neither SHE1 nor IVR mRNA was detected by RT-PCR, with or without TMV infection (Figure 5C). These results suggest that SHE1 affects the accumulation of IVR and that SHE1 may be a TF in the pathway for synthesis of IVR. Alternatively, SHE1 may interact with an upstream component of the pathway, leading to IVR, and this interaction is required for stimulating production of IVR.

### 3.4. Interaction between SHE1 and IVR

A classical feature of many biosynthetic pathways is “feedback inhibition”, in which the end-product interacts with either an early or intermediate factor to prevent further biosynthesis. To determine whether the TF SHE1 interacts with IVR, the yeast two-hybrid (Y2H) system was used, in the first instance, with *SHE1* in the pAS2 vector (bait) and *IVR* in the pACT2 vector (prey). This assay showed that SHE1 interacted strongly with IVR and that neither SHE1 nor IVR alone autoactivated the expression of the reporter genes (Figure 6; Appendix A).

To confirm this result, a co-immunoprecipitation assay was used, in which SHE1 fused to a red fluorescent protein (SHE1-RFP; to better distinguish the electrophoretic mobility of SHE1 from IVR) was co-expressed with IVR fused to an HA-tag (IVR-HA) in *N. benthamiana* leaves, and extracts of such leaves were co-immunoprecipitated using an anti-HA tag antibody. The immunoprecipitated material was analyzed by SDS-PAGE and Western blotting. The Western blots were probed with either anti-IVR antiserum to detect the IVR-HA or anti-SHE1 antiserum to detect the SHE1-RFP (Figure 7A). The results showed that SHE1-RFP could be neither precipitated nor detected by the anti-IVR antibody, whereas IVR-HA could not be detected by the anti-SHE1 antibody. However, both proteins could be detected in the co-precipitated material (Figure 7A). Thus, SHE1 interacts with IVR, supporting the view that SHE1 is a TF in the IVR biosynthetic pathway.

To delimit the region of SHE1 interacting with IVR, the Y2H system was deployed to assess the interactions of three partial C-terminal deletions of SHE1 with the full-length IVR and of two N-terminal partial deletions of IVR with the full-length SHE1 (Figure 6); the choice of deletion constructs being determined by (putative) functional domains in SHE1 and IVR (see Discussion). The Y2H results indicated that only the C-terminal half of IVR (~300 nt, corresponding to 99 amino acids) was required for the interaction of SHE1 and IVR (Figure 6). Neither of the two partial IVR deletion constructs alone, nor any of the three partial SHE1 constructs alone, was able to autoactivate the expression of the reporter genes in the absence of partner protein (Appendix A). By contrast, the N-terminal 79% of SHE1 (555 nt of 690 nt; 185 amino acids) was required for strong binding to IVR; however, the interaction was weaker in the Y2H assays using a construct containing a further deletion towards the N-terminus (405 nt, retaining the N-terminal 58% of SHE1; 135 amino acids), and essentially was lost using an additional deletion construct (255 nt, retaining the N-terminal 37%; 85 amino acids) (Figure 6). This suggests that SHE1 sequences between amino acids 86 and 185 were essential for interaction of SHE1 with amino acids 101–199 of IVR, but a weak interaction still occurred when only the N-terminal 135 amino acids of SHE1 were present.

### 3.5. Co-Localization of SHE1 and IVR

To further verify the interactions between SHE1 and IVR, a BiFC assay was used, in which sequences expressing SHE1 and IVR were fused to sequences expressing different halves of a modified, enhanced YFP in the corresponding vectors [28]. BiFC also allowed the determination of the subcellular distribution of the fusion proteins. The BiFC assay results showed that SHE1 interacted with IVR and suggested that the YFP signal was distributed in the appressed cytoplasm and on the tonoplast membrane (Figure 7B).

To confirm the co-localization of the SHE1 and IVR proteins in fixed protoplasts, we used the Duolink in situ PLA, which uses primary antibodies to two interacting components, amplification of the signal and a fluorescent dye (Tx Red) to detect their proximity. This system showed the SHE1-HA and IVR-FLAG interacted and produced a red signal throughout most of the cytoplasm (Figure 7C). Thus, SHE1 interacts with IVR and is found located throughout the cytoplasm and on tonoplast membranes.

## 4. Discussion

The induction of *SHE1* by four viruses, the accumulation of which was not affected at 3 dpi in the infected leaf, indicates that induction of *SHE1* does not require the *N* gene, although the activation of the *N* gene may greatly increase the level of induction (Figure 1). By contrast, the restriction of infection by TMV-GFP was determined to require the *N* gene, since it did not occur in either tobacco cv. Samsun NN or tobacco cv. SR1 rendered resistant to WT TMV by transgenically introducing the *N* gene [25]. However, this restriction response is mediated through effects on TMV-GFP movement [25], even at a temperature at which SHE1 is not expressed [15]. Thus, SHE1 is not a component of the novel resistance response in *N* gene tobacco plants, preventing the systemic infection of TMV-GFP.

The silencing or over-expression of *SHE1* had different effects on local infection by WT TMV, with over-expression of *SHE1* showing a reduction in virus local movement, as well as the number of local lesions, whereas silencing of *SHE1* had no notable effects on either phenotype (Figure 2). Curiously, there was also a change in the physical nature of the lesion on the OEx-SHE1 line, with the lesion being dark brown in appearance (Figure 2A). This is similar to the TMV-induced local lesions seen in leaf disks of both nontransformed and transgenic SNN tobacco plants silenced for the expression of both kinases WIPK and SIPK, while soaking in methyl jasmonate [34]. However, in that case, the lesions were larger with the phytohormone treatment (Figure 5B in [34]), whereas here, the lesions were smaller in the overexpressing plants than in the control. 

In the case of the effects on local and systemic movement of TMV-GFP at the non-restrictive temperature (for WT TMV), over-expression of *SHE1* led to less intense fluorescence in the inoculated leaves than in the nontransformed tobacco, and no systemic infection, as also was the case for the control (Figure 3). However, silencing *SHE1* had no effect on local movement of TMV-GFP, with no systemic movement in the first 2 wpi (Figure 3). At later times, WT TMV showed systemic movement at the restrictive temperature for plants silenced for *SHE1* expression (Figure 4). Interestingly, TMV also was able to infect systemically the WIKP/SIPK doubly silenced transgenic plants, showing necrotic local lesions in distal leaves [34]. This indicates a perturbation in systemic virus resistance by altering the expression of other genes involved in the signaling defense response, as also was seen here (Figure 4). By contrast, over-expression of *SHE1* reduced local accumulation and inhibited systemic infection, as also noted by Fischer and Dröge-Laser [15], although the latter effect could be a consequence of the former effect. The slow systemic movement that occurred when SHE1 expression was blocked by silencing during infection by WT TMV (Figure 4) may reflect non-vascular, systemic cell-to-cell movement, as seen in SNN tobacco plants by TMV mutants in the absence of functional capsid protein [35]. In that case, a slow cell-to-cell movement of the virus occurred to parenchyma cells in the petiole and up the parenchyma cells of the stem, exiting into parenchyma cells of an upper leaf petiole, and moving into mesophyll cells of a partially expanded upper leaf [35]. These effects are reminiscent of the delayed and incomplete systemic infection of transgenic SNN tobacco expressing the bacterial *NahG* gene, encoding salicylate dehydroxylase and inhibiting the accumulation of SA [36]. Given that there are multiple events in the resistance against TMV in SNN tobacco, involving components such as RDR1, AOX1, pathogenesis-related proteins and IVR [1], the loss of one pathway may partially compromise the overall resistance, but does not abolish it.

Transiently expressed SHE1 fused to YFP was found to localize in the nucleus, but when interacting with the CMV 1a protein, SHE1 was found accumulating in the nucleus but also on the tonoplast membrane [17]. The ability of SHE1 to interact with IVR was shown using different approaches (Figure 6 and Figure 7), and the co-localization of the interacting SHE1-IVR complexes with the cytoplasm and tonoplast membrane (Figure 7B,C) suggests that IVR accumulation results in sequestering SHE1 from the nucleus, reducing further *IVR* gene expression. The co-regulation of the expression of *SHE1* and *IVR* in *SHE1* over-expressing and silenced plants (Figure 5; Appendix A) together suggest that IVR is in the same biosynthetic pathway downstream from SHE1. Thus, the binding of IVR to SHE1 to prevent its TF function would an example of end-product feedback inhibition. It is not known whether SHE1 itself binds to the uncharacterized promoter region of *IVR*, or whether it binds to another promoter for a gene encoding the actual TF for IVR mRNA synthesis.

A further aspect of the relationship between IVR and SHE1 can be discerned by the inhibitory effect of high temperature on the expression of *SHE1* [15] and *IVR*, as well as the function of IVR. IVR was found to be undetectable in either the culture fluid of SNN tobacco protoplasts infected with TMV or leaves of SNN tobacco infected with TMV, both maintained at 35 °C, using either serology or bioassay [37]. Interestingly, an amphidiploid interspecific *N*. *glutinosa* × *N*. *debneyi* hybrid line that showed a constitutive expression of IVR protein, giving rise to very few and smaller lesions when infected by TMV [38], and losing both IVR accumulation and resistance to TMV when maintained at 35 °C [37].

The region of SHE1 that was delimited for interaction with IVR (Figure 6) constituted amino acids 86-185 of SHE1 (blue amino acids in Figure 8), although either these sequences or those from amino acids 1-86 could be involved with the proper folding required for the other region to interact with IVR. Nevertheless, this domain overlaps with the conserved DNA binding domain present in AP2/ERF TFs (blue underlined amino acids in Figure 8) [15]. A further deletion from amino acid 185 down to 135 weakened the binding ability of SHE1 for IVR. This region contains most of the amphipathic α–helix from the SHE1 protein (bold blue sequences in Figure 8). Thus, part of the C-terminal half of IVR may be interacting with the SHE1 DNA-binding domain and a key structural feature of this domain, preventing SHE1 from functioning as a TF. The SHE1 deletion construct missing ~20% of its C-terminal region was not as strong at binding IVR as the full-length SHE1 construct (Figure 6). This could be due to some structural alteration in part of the remaining protein, caused by the deletion, or it could be due to the sequences immediately adjacent to amino acid 185 also binding to the IVR. These adjacent SHE1 sequences (amino acids 186–203) contain a potential nuclear localization signal (NLS). Thus, if IVR binds to both the NLS and the DNA binding domain of SHE1, then the bound SHE1 would not be able to enter the nucleus.

The published sequence of tobacco SNN IVR [18] has some errors in the C-terminal proximal coding region (Appendix A). These errors were noticed when sequencing the cDNA clone of IVR provided by [18]. In addition, the revised sequence is now nearly identical for this region with IVR sequences in the tobacco genome sequence, as well as *IVR*-like gene sequences identified in sequenced mRNAs from potato, tomato, pepper, Arabidopsis, rice and wheat [39]. This confirms that the original tobacco IVR cDNA sequence contained three errors: (1) an extra G shown as nucleotide 509; (2) a single rather than a double C present at nucleotide 542; and (3) the TC pair at nucleotides 575–576 (Figure 2 in [18]) should be CT (Appendix A). These errors led to a 12-amino-acid frameshift: 25-36 amino acids from the C-terminus of the 199-amino-acid IVR protein (Appendix A). Nevertheless, the statement by Akad et al. [18] that the C-terminal 120-199-amino-acid region is highly acidic (now containing 21 aspartate and glutamate residues), hydrophobic and containing a helical structure, is still the case. This also is the same region that contains sequences required for interaction with SHE1 (Figure 6). We presume that the level of IVR would be considerably higher than the level of a TF involved in IVR production, which would also facilitate feedback inhibition. However, after about a week, IVR was shown to accumulate in the intercellular spaces of infected SNN tobacco leaves [22]. In addition, this extracellular IVR, as well as the IVR that accumulated in the culture medium of TMV-transfected SNN tobacco protoplasts, between 24 and 96 hpi [19], were identical in electrophoretic movement, which was different from IVR expressed in *E. coli* [18]. This suggests that the in planta expressed IVR was post-translationally modified. That modification could also have some effect on IVR binding to SHE1. 

## Figures and Tables

**Figure 1 viruses-15-00059-f001:**
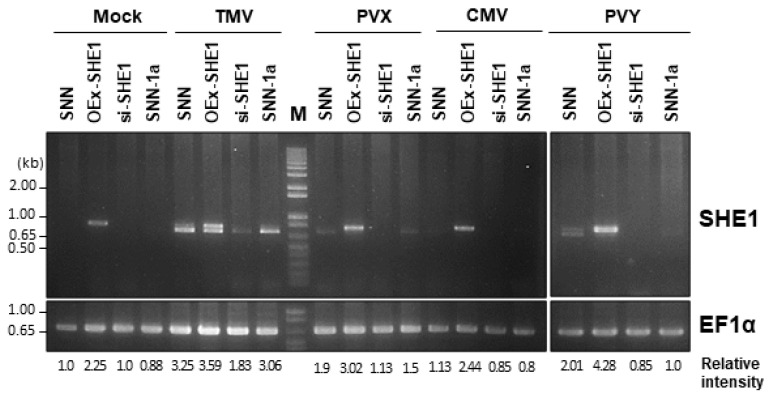
Semi-quantitative RT-PCR for levels of SHE1 mRNA accumulation, at 3 days post inoculation. Plants were either nontransformed tobacco cv. Samsun NN (SNN) or transgenic tobacco plants generated for over-expression of SHE1 (OEx-SHE1), silencing of expression of SHE1 (si-SHE1) or expression of the CMV 1a protein (SNN-1a). Plants were either mock inoculated (Mock) or inoculated with TMV, PVX, CMV or PVY. The RT-PCR reactions used EF1α as a reference gene. M = 1 kb plus DNA Ladder (Invitrogen, Carlsbad, CA, USA). Since 35 cycles were found to be saturating, 25 cycles were chosen for the PCRs [17].

**Figure 2 viruses-15-00059-f002:**
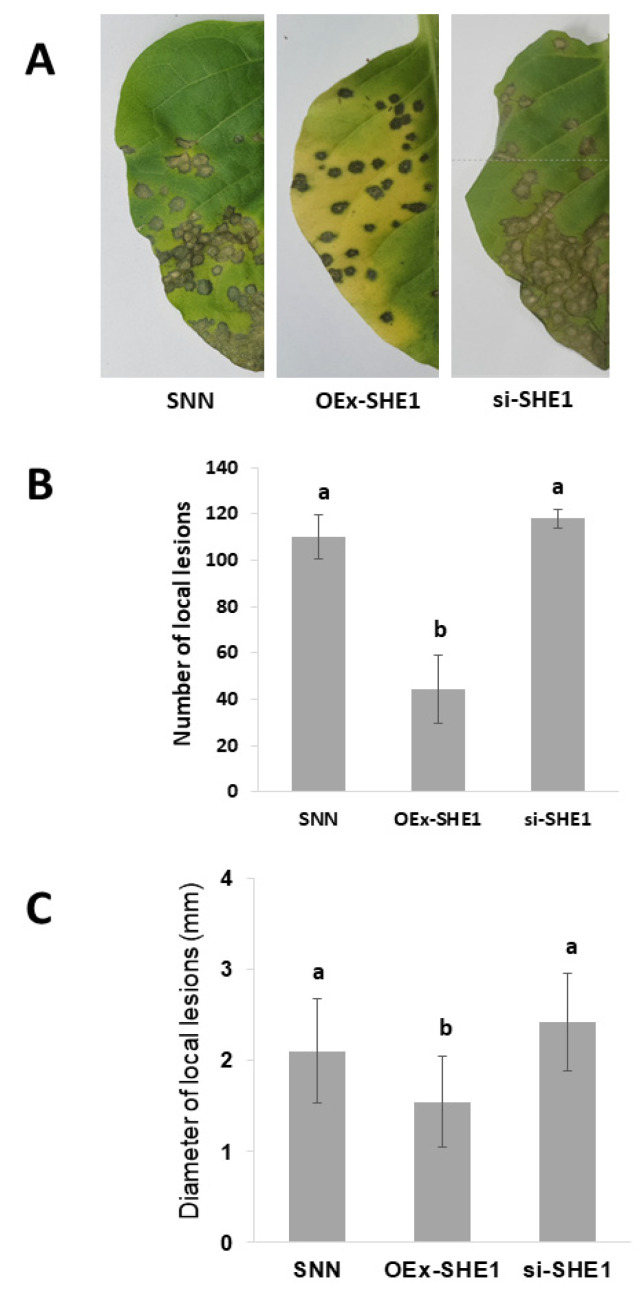
Effect of silencing or over-expressing SHE1 on local movement of TMV in infected tobacco cv. Samsun NN. (**A**) Local lesion induced on half-leaves of tobacco plants inoculated with TMV, nontransformed (SNN) or either transformed to overexpress (OEx-SHE1) or silence (si-SHE1) the SHE1 gene. (B, C) Histograms showing the mean and standard deviation (error bars) of either (**B**) the number of local lesions or (**C**) the estimated diameter of local lesions induced by TMV infection on SNN, SNN-OEx-SHE1 or SNN-si-SHE1. The data were analyzed statistically by Duncan’s multiple range test (DMRT). Different letters above the error bars indicate a significant difference at *p* < 0.01.

**Figure 3 viruses-15-00059-f003:**
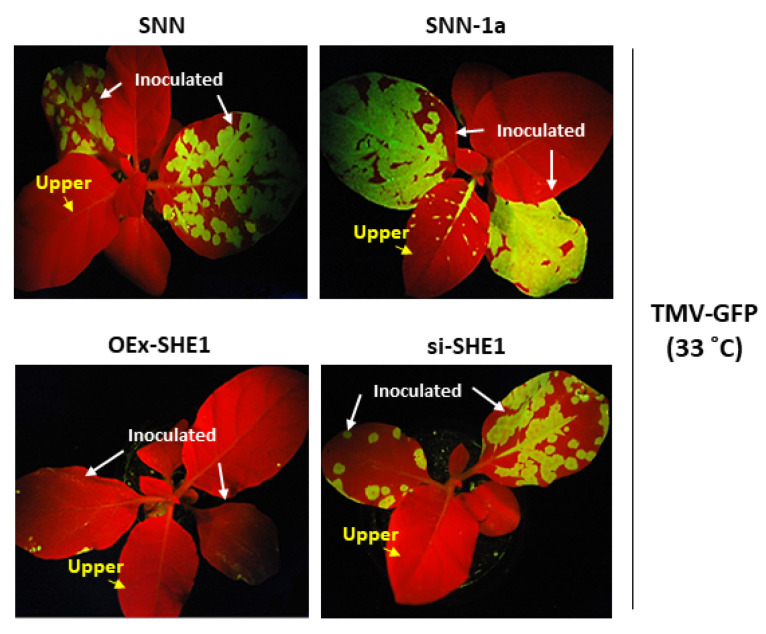
Effect of silencing or over-expressing SHE1, or expression of the CMV 1a protein on systemic movement of TMV-GFP at 33 °C. Plants were transgenic for expression of CMV 1a (SNN-1a), overexpression of SHE1 (OEx-SHE1) or silencing of expression of SHE1 (si-SHE1). Plants were inoculated with TMV-GFP and maintained at 33 °C for 2 weeks post inoculation. Whole plants were photographed under UV light. OEx-SHE1 line #5 and si-SHE1 #4 were used for the experiments. The inoculated and upper leaves are indicated by white and yellow arrows, respectively.

**Figure 4 viruses-15-00059-f004:**
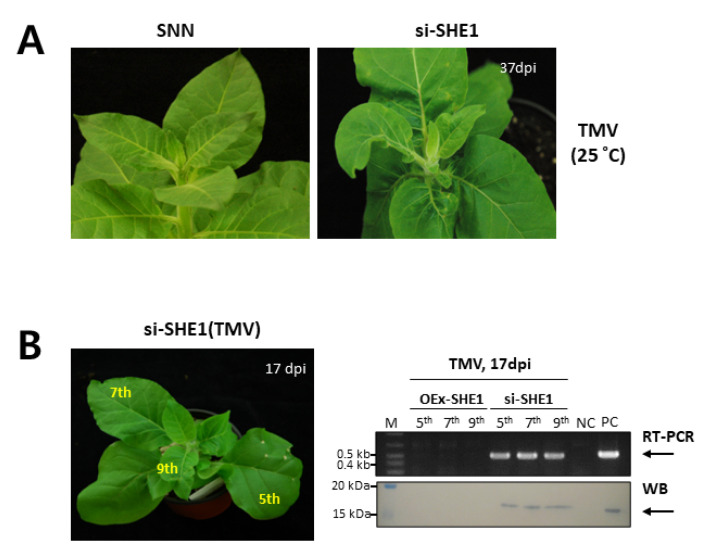
Systemic infection of nontransformed and transgenic plants silenced for expression of SHE1 inoculated with TMV. (**A**) Transgenic tobacco cv. Samsun NN si-SHE1 plants inoculated with TMV were maintained for the time indicated for 37 dpi. (**B**) Assay for TMV accumulation in the 17 dpi infected plant on the left, by either RT-PCR for TMV RNA (*CP* gene; upper right) or Western blot analysis for TMV coat protein (lower right). Black arrows indicate the position of the expected PCR-sized product (523 bp) and the virus CP. TMV-infected plants were photographed under daylight conditions.

**Figure 5 viruses-15-00059-f005:**
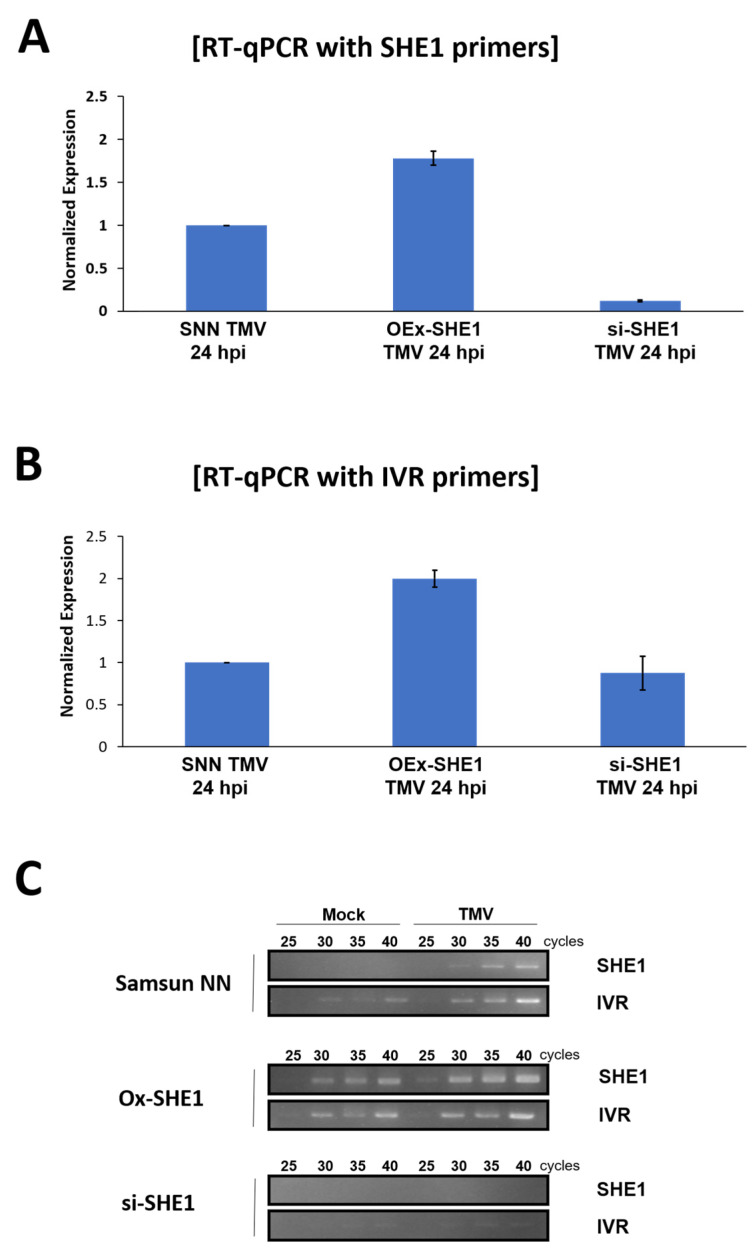
Analysis of SHE1 and IVR mRNA accumulation in nontransformed (SNN) and transgenic Samsun NN tobacco plants. Plants were transgenic for either overexpression of SHE1 (OEx-SHE1) or silencing of expression of SHE1 (si-SHE1). Plants were inoculated with TMV. Total RNAs from healthy and inoculated leaves were extracted and processed at 24 h post inoculation, followed by either RT-qPCR analysis (**A** and **B**) or semi-quantitative RT-PCR analysis (**C**), for the level of (**A**) SHE1 mRNA, (**B**) IVR mRNA or (**C**) both. The mRNA from eIF1α was used as a reference standard.

**Figure 6 viruses-15-00059-f006:**
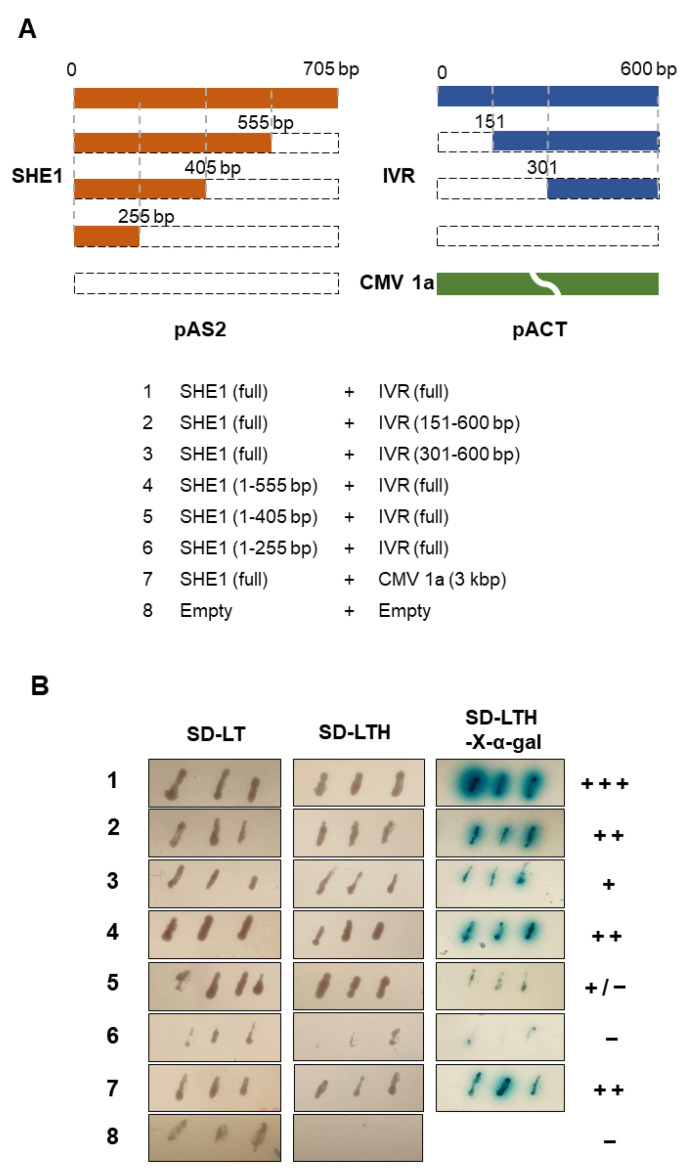
Yeast two-hybrid assays for interaction of SHE1 with IVR. (**A**) Schematic presentation of full-length or partial C-terminally deleted *SHE1* genes (in the pAS2 vector) and full-length or partial N-terminally deleted *IVR* genes (in the pACT2 vector) used to co-transform into yeast cells. (**B**) SD-L, SD-T, SD-LT and SD-LTH represent assays in which transformants were selected on SD/-Leu, SD/-Trp, SD/-Leu/-Trp and SD/-Leu/-Trp/-His plates without X-α-gal, to detect the initiation of transcription of amino acid biosynthesis reporter genes (LEU, TRP and HIS). The X-α-gal assay represents colonies transferred to filters and assayed for reaction with X-α-gal; + = positive interaction; – = negative interaction. SHE1 in pAS2 and CMV 1a in pACT were co-transformed as a positive control.

**Figure 7 viruses-15-00059-f007:**
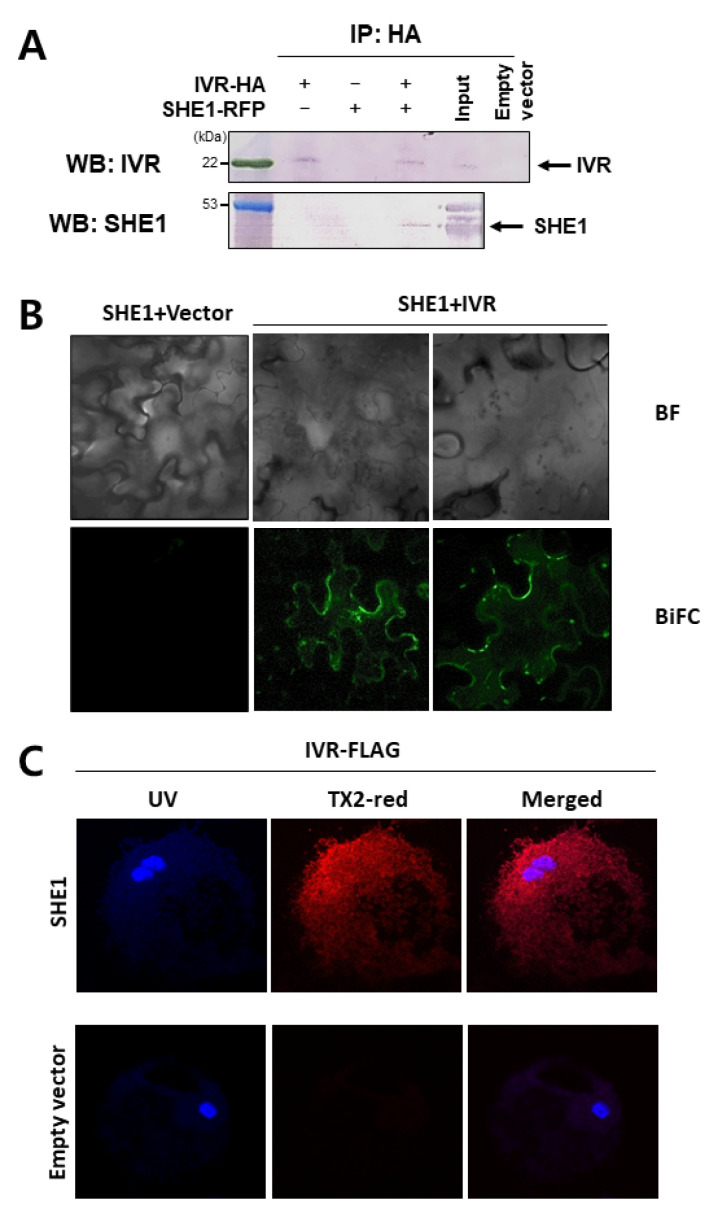
Interactions between IVR and SHE1. (**A**) Co-IP assay showing interaction of IVR and SHE1 in vivo. *N. benthamiana* plants were infiltrated with *Agrobacterium* harboring plasmids expressing IVR tagged with HA (IVR-HA), SHE1 fused to the red fluorescent protein (SHE1-RFP), both plasmids, or an empty vector control. At 3 days after infiltration, the agroinfiltrated leaves were extracted and subjected to immunoprecipitation (IP) performed with antiserum to the HA-tag on IVR. The immunoprecipitated material was denatured and fractionated by SDS-PAGE and electroblotted to nitrocellulose membranes. Western blot detection was performed using antisera to IVR or SHE1. M = Protein molecular weight markers. (**B**) Co-localization of SHE1 and IVR in *N. benthamiana* as determined by bimolecular fluorescence complementation (BiFC) of SHE1 and IVR fused to halves of enhanced YFP (eYFP) expressed in agroinfiltrated *N. benthamiana* leaves and examined by confocal microscopy. The agroinfiltrated plasmids control shown contained SHE1 fused to the N-terminal half of eYFP and the empty vector for expressing the C-terminal half of eYFP. Upper panels show the bright field (BF) view, whereas lower panels show the BiFC, viewed under UV light. (**C**) Co-localization of SHE1 and IVR in *N. benthamiana* as determined by Duolink in situ proximity ligation assay in transfected protoplasts co-expressing IVR-FLAG and SHE1-HA, detected by mouse anti-FLAG mAb and rabbit anti-HA mAb, respectively, with co-localization detected by Texas red dye (TX2-red) using confocal microscopy (upper panels). The nucleus was detected by DAPI staining (blue). The control transfection, expressing IVR-FLAG plus an empty HA-vector, is shown in the lower panels. Note: the presence of fixed cytoplasmic constituents above the nucleus, containing proteins interacting with each other and detected by TX2-red in the Duolink detection system, means that the red dye above the nucleus does not allow one to distinguish interactions that occur in the cytoplasm alone from those occurring in both the cytoplasm and the nucleus.

**Figure 8 viruses-15-00059-f008:**
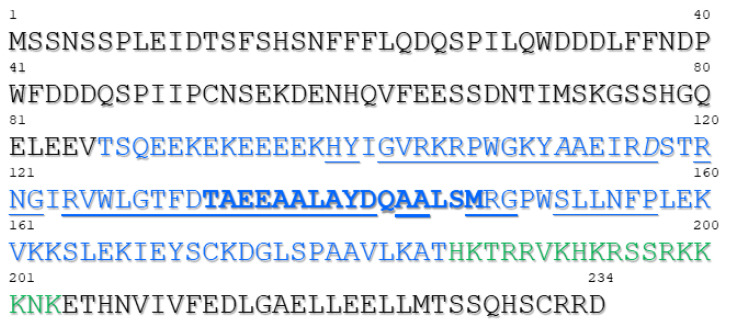
Amino acid sequence of tobacco SHE1, with proposed functional domains indicated. The SHE1–IVR binding region (amino acids 86-185) is shown in blue letters. Identical or highly conserved amino acids between AP2/ERFs transcription factors (TFs) and related proteins, corresponding to DNA-binding domain found in AP2/ERF TFs, are underlined (amino acids 99–157). Italicized blue residues (A112 and D117) are required for GCC binding. An amphipathic α-helix at amino acids 132–147 is shown in bold blue. The basic region at amino acids 186–203 is a proposed nuclear localization signal, shown in green. The amino acids at the N and C termini resemble those shown to be involved in transcription activation. The N and C termini are both acidic.

## Data Availability

The data presented in this study are available in the article and Appendix A.

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
