# Peer review of "The Virus-Induced Transcription Factor SHE1 Interacts with and Regulates Expression of the Inhibitor of Virus Replication (IVR) in N Gene Tobacco"

_viruses, 2022, doi:10.3390/v15010059_

Round 1
Reviewer 1 Report
Review of Yoon et al.
In this manuscript Yoon and collaborators investigated relations between transcription factor SHE1 and an inhibitor of viral replication (IVR) in tobacco plants, and their effect/response to viral infections
They show first that infection by some RNA viruses, in particular by TMV, increases the mRNA levels of SHE1 in four types of Samsun NN tobacco plants: non-transformed, transgenic expressing the 1a protein of Fny CMV, transgenic over-expressing SHE1 (from tobacco Xanthi nc.) or transgenic expressing inverted repeat SHE1 fragments, and thus having partially silenced the endogenous gene. Authors show that transgenic overexpression of SHE1 can have a detrimental effect on TMV (or TMV-GFP) local accumulation and systemic movement in the temperature-dependent, NN plant background
Second, they show that TMV-induced changes in the levels of SHE1 mRNA in transgenic plants over-expressing or silencing SHE1 coincides with similar variations in the expression of IVR (inhibitor of viral replication) mRNA, suggesting a common signaling pathway
Finally authors show that IVR and SHE1 variants interact using Y2H, as well as in planta using different techniques (pull down, BiFC, Duolink), and N. benthamiana cells transiently-expressing those variants
I think that the observations and the conclusions are interesting, and sustained by the experimental data presented
Overall the paper is well written. However, I think that the way in which Results are presented need improvements. There are also some other small issues in the paper that authors should consider:
Title:
I think that a bracket is missing (IVR)
Methods:
They are overall well written. However, some data are not clear. With regard to the semi-quantitative and quantitative RT-PCRs, the oligos used are said to be indicated in Table S2. But the SHE and IVR fragments that they amplify are 700 and 600 nt, hardly adequate for qPCR. Please make sure that the list is correct
Results:
In Figure 1 there are two bands in the semi-quantitative RT-PCR amplification of SHE1 mRNA fragments. It appears that the PCR product from the over-expressed transgene is larger than that from the endogenous gene. This is a guess by the reader, until it is confirmed much further down the manuscript in paragraph 295-303, after Figures 5 and S5. Obviously this paragraph should appear immediately after Figure 1
Figure 5. Charts: the X axis lane is absent from the left chart. I do not understand the need or reason to show the amplification runs of the RT-qPCRs that appear below the charts. I understand that the level of SHE1 mRNA in non-transformed tobacco has been given the arbitrary value of 1. However, in the IVR qPCR chart, the value of IVR mRNA in the same type plant is not 1. Why? In addition, in the charts there no error bars or significance analysis. Please amend this Figure.
I think that Figure S5 should appear in the main text because it is mentioned several times in the Results section. Either as it is now, or in chart form. Its data reinforces and complements results shown in Figure 5. For example, the result commented in the sentence in line183-284 is not shown in Figure 5, despite what is said in that sentence.
Figure 7 would benefit from better labeling. The legend is very long. In 7C, please indicate with arrows the protoplast nuclei (N, blue?) and sites of SHE1/IVR interaction (violet?)
Author Response
Answers to queries/suggestions made by Reviewer 1
Thank you very much for your valuable suggestions/comments from the reviewer. The queries are improved our research article quality. Care has been taken to incorporate all possible suggestions made by reviewer in the revised version of the manuscript to improve the scientific quality of the manuscript.
Reviewer 1.
In this manuscript Yoon and collaborators investigated relations between transcription factor SHE1 and an inhibitor of viral replication (IVR) in tobacco plants, and their effect/response to viral infections
They show first that infection by some RNA viruses, in particular by TMV, increases the mRNA levels of SHE1 in four types of Samsun NN tobacco plants: non-transformed, transgenic expressing the 1a protein of Fny CMV, transgenic over-expressing SHE1 (from tobacco Xanthi nc.) or transgenic expressing inverted repeat SHE1 fragments, and thus having partially silenced the endogenous gene. Authors show that transgenic overexpression of SHE1 can have a detrimental effect on TMV (or TMV-GFP) local accumulation and systemic movement in the temperature-dependent, NN plant background
Second, they show that TMV-induced changes in the levels of SHE1 mRNA in transgenic plants over-expressing or silencing SHE1 coincides with similar variations in the expression of IVR (inhibitor of viral replication) mRNA, suggesting a common signaling pathway
Finally authors show that IVR and SHE1 variants interact using Y2H, as well as in plant using different techniques (pull down, BiFC, Duolink), and N. benthamiana cells transiently-expressing those variants
I think that the observations and the conclusions are interesting, and sustained by the experimental data presented
Overall the paper is well written. However, I think that the way in which Results are presented need improvements. There are also some other small issues in the paper that authors should consider:
Title: I think that a bracket is missing (IVR)
Response: Corrected. Thank you.
Methods:
They are overall well written. However, some data are not clear. With regard to the semi-quantitative and quantitative RT-PCRs, the oligos used are said to be indicated in Table S2. But the SHE and IVR fragments that they amplify are 700 and 600 nt, hardly adequate for qPCR. Please make sure that the list is correct.
Response: As stated on Page 3, line 121, We performed semi-quantitative RT-PCR with the primers listed in Table S2 and the result is presented in Fig. 1. Similarly, as stated on Page 4, line 125, For quantitative PCR, we used the primers described by Baek et al., 2017.
Results:
In Figure 1 there are two bands in the semi-quantitative RT-PCR amplification of SHE1 mRNA fragments. It appears that the PCR product from the over-expressed transgene is larger than that from the endogenous gene. This is a guess by the reader, until it is confirmed much further down the manuscript in paragraph 295-303, after Figures 5 and S5. Obviously this paragraph should appear immediately after Figure 1
Response: The reviewer has a point. We placed the explanation of the double bands for SHE1 in the OEx-SHE1 plants where it was, because it also applied to Figure S5. [All Figures (and Tables) must be called out in numerical order, and therefore we could not refer to this being in Figure S5 until after we mentioned Figure S4.] Therefore, here, we have moved the explanation to after Figure 1, but have deleted mention of Figure S5 with that text.
Figure 5. Charts: the X axis lane is absent from the left chart. I do not understand the need or reason to show the amplification runs of the RT-qPCRs that appear below the charts. I understand that the level of SHE1 mRNA in non-transformed tobacco has been given the arbitrary value of 1. However, in the IVR qPCR chart, the value of IVR mRNA in the same type plant is not 1. Why? In addition, in the charts there no error bars or significance analysis. Please amend this Figure.
Response: Figure 5 was amended to include error bars as well as a add a title for the y-axis (“Normalized Expression”). [Note: The x-axis is the horizontal one; the y-axis is the vertical one.] The reviewer asks why the scale for IVR is so much lower and there is no. "1" level for this. This is because all things are set to the level of SHE1 in TMV-infected SNN plants at 24 hpi as 1, including the IVR samples. If you look at the RT-qPCR curves for IVR, it is also clear that there are multiple curves, but they are not as separated from each other (into three groups) as for the SHE1 curves, because the scale is much smaller (around 0.015 to 0.03). The curves also show a mean between 30 and 35 cycles, while for SHE1, even for the lowest level (si-SHE1) the mean is at 30 cycles. This is because, at 24 hpi, in TMV infected SNN plants, SHE1 mRNA is at the maximum or already starting to go down (varies in different experiments) while IVR mRNA in TMV infected SNN is just rising. It has not reached its maximum yet.
I think that Figure S5 should appear in the main text because it is mentioned several times in the Results section. Either as it is now, or in chart form. Its data reinforces and complements results shown in Figure 5. For example, the result commented in the sentence in line183-284 is not shown in Figure 5, despite what is said in that sentence.
Response: Based on comments by the reviewer, we also added the semi-quantitative RT-PCR data to Figure 5 and moved the cross-threshold curves to Figure S5.
Figure 7 would benefit from better labeling. The legend is very long. In 7C, please indicate with arrows the protoplast nuclei (N, blue?) and sites of SHE1/IVR interaction (violet?)
Response: On pg 13, ln 380, it is stated that the nucleus is stained blue by DAPI. On pg 14, ln 413-415, it was stated that the detection of the interaction was done using a fluorescent dye (Tx-Red), and that the red signal was located throughout the cytoplasm. Therefore, the red area is the site of interaction for SHE1-IVR.

Reviewer 2 Report
The MS entitled “ The Virus-induced Transcription Factor SHE1 Interacts with and Regulates Expression of the Inhibitor of Virus Replication (IVR in N Gene Tobacco” explores the interaction of SHE1 and IVR during TMV infection. The authors developed SHE1 overexpression and RNAi lines and studied the response of transgenic lines in TMV infection and how TMV infection regulates SHE1 and IVR expression and interaction. Later, they have shown the physical interaction between SHE1 and IVR using Y2H, BiFC and proximity ligation assay and provided molecular evidence of interaction by immunoprecipitation.
This MS provides evidence for role of SHE1 and IVR during TMV infection which could be helpful to develop antivirus strategies against TMV. I recommend accepting this MS after following comments are resolved.
I have following comments on this MS.
Minor comments:
The introduction part could be more concise. There are few lengthy sentences which could be rephrased.
In the MS title the bracket is missing for IVR.
Major comments:
Why the infection of SHE1 is downregulated in OEX-SHE1 with TMV-GFP infection (Fig S3), while in Fig 5 authors showed that TMV infection induced the expression of SHE1 in OEX-SHE1 line using qRT-PCR?
In FigS5 SHE1 transcript could not be detected by semi-quantitative RT-PCR in wild type SNN plants, while the northern blot data suggest that SHE1 is expressed in SNN plants. The northern blot data contradict the Sq-RTPCR data as PCR is considered to be more sensitive than northern hybridization.
The authors stated the reason for double bands shown in Sq-RTPCR in Fig 1 is due to the difference in length of the SHE1 transcript coming from two different sources. The difference in PCR fragments is 9 nucleotides. Differentiating two DNA fragments (700bp long) with 9nt difference on agarose gel is very challenging. Could author please provide details of agarose gel percentages used to resolve these fragments?
In figure 5A and 5B the graphs don’t have error bars? Was this experiment repeated only once? Authors are suggested to either delete the melt-curve data or move it to supplementary data.

Author Response
Answers to queries/suggestions made by Reviewer 2
Thank you very much for your valuable suggestions/comments from the reviewer. The queries are improved our research article quality. Care has been taken to incorporate all possible suggestions made by reviewer in the revised version of the manuscript to improve the scientific quality of the manuscript.
Reviewer 2.
The MS entitled “ The Virus-induced Transcription Factor SHE1 Interacts with and Regulates Expression of the Inhibitor of Virus Replication (IVR in N Gene Tobacco” explores the interaction of SHE1 and IVR during TMV infection. The authors developed SHE1 overexpression and RNAi lines and studied the response of transgenic lines in TMV infection and how TMV infection regulates SHE1 and IVR expression and interaction. Later, they have shown the physical interaction between SHE1 and IVR using Y2H, BiFC and proximity ligation assay and provided molecular evidence of interaction by immunoprecipitation.
This MS provides evidence for role of SHE1 and IVR during TMV infection which could be helpful to develop antivirus strategies against TMV. I recommend accepting this MS after following comments are resolved.
I have following comments on this MS.
Minor comments:
The introduction part could be more concise. There are few lengthy sentences which could be rephrased.
Response: It would help if the reviewer could specify which sentences.
In the MS title the bracket is missing for IVR.
Response: This has been corrected. Thank you.
Major comments:
Why the infection of SHE1 is downregulated in OEX-SHE1 with TMV-GFP infection (Fig S3), while in Fig 5 authors showed that TMV infection induced the expression of SHE1 in OEX-SHE1 line using qRT-PCR?
Response: We do not know why the level of SHE1 was reduced by TMV-GFP infection in OEx-SHE1 plants in Figure S3. In Figure 5, TMV infection did increase the level of SHE1 expression in OEx-SHE1 plants, but as shown if the movement experiments in Figures 3 vs. 4, you cannot directly compare effects of TMV vs. TMV-GFP, since TMV-GFP does not overcome other aspects of the defense response that TMV overcomes.
In FigS5 SHE1 transcript could not be detected by semi-quantitative RT-PCR in wild type SNN plants, while the northern blot data suggest that SHE1 is expressed in SNN plants. The northern blot data contradict the Sq-RTPCR data as PCR is considered to be more sensitive than northern hybridization.
Response: Interesting point. The same is true in Figure 1, here there is no SHE1 detectable in the mock-inoculated SNN plants. It may be that the level is so low that it would take more than 40 cycles of PCR to see it in non-inoculated plants. On the other hand, a northern blot detects what it present and depends on the level, the specific activity of the probe, and the length of exposure, while in the case of RT-PCR, after the RT step the sample must be diluted considerably before PCR to prevent inhibition of PCR by the high total nucleic acid concentration present from the previous step.
The authors stated the reason for double bands shown in Sq-RTPCR in Fig 1 is due to the difference in length of the SHE1 transcript coming from two different sources. The difference in PCR fragments is 9 nucleotides. Differentiating two DNA fragments (700bp long) with 9nt difference on agarose gel is very challenging. Could author please provide details of agarose gel percentages used to resolve these fragments?
Response: The agarose gel percent was 1.2 %. It should be noted that the terminator of the SNN SHE1 is 21 nt farther than the terminator of the Xanthi-nc SHE1. This has been clarified in the explanation.
In figure 5A and 5B the graphs don’t have error bars? Was this experiment repeated only once? Authors are suggested to either delete the melt-curve data or move it to supplementary data.
Response: No, the experiments were repeated. The error bars have been added, and the quantitative PCR curves (showing the crossover thresholds) – not melting curves – have been placed into Figure S5.

Reviewer 3 Report
The inhibitor of virus replication (IVR) is a plant factor discovered many years ago by Loebenstein and colleagues that is associated with virus resistance in tobacco. This research group have investigated a transcription factor SHE1, which they show in this paper to regulate expression of IVR. In previous work they showed that SHE1 interacts with the 1a protein of cucumber mosaic virus (CMV). This was significant since transgenic expression of the 1a protein in tobacco plants inhibits viruses resistance. In this work they show that TMV infection of Samsun NN tobacco (which results in a hypersensitive response, HR) triggers increased SHE transcription accumulation. PVY, CMV and PVX caused only modest increases in the SHE1 transcript signal. Presumably because these viruses do not trigger the string HR resistance mechanism in Samsun NN. Engineered increased or decreased expression of SHE1 alters the size, number and morphology of TMV induced HR lesions on Samsun NN which increased expression enhancing resistance and decreased expression diminishing resistance to the virus. Examining TMV-GFP systemic movement in Samsun NN with increased/decreased SHE1 expression above the critical temperature range for N gene mediated resistance confirmed the role of SHE1 in resistance. At lower temperatures, decreasing the SHE1 level enhances virus spread. They confirm using Y2H, Co-IP and cell biology that SHE1 and IVR interact, suggesting an interaction with regulatory importance, and identify putative functional domains in the SHE1 protein primary sequence. The work is well carried out , the paper is well written and the findings are important for our understanding of plant virus resistance.
Minor corrections
Title
... (IVR) in N Gene....
line 273 temperature indicated?
Author Response
Answers to queries/suggestions made by Reviewer 3
Thank you very much for your valuable suggestions/comments from the reviewer. The queries are improved our research article quality. Care has been taken to incorporate all possible suggestions made by reviewer in the revised version of the manuscript to improve the scientific quality of the manuscript.
Reviewer 3.
The inhibitor of virus replication (IVR) is a plant factor discovered many years ago by Loebenstein and colleagues that is associated with virus resistance in tobacco. This research group have investigated a transcription factor SHE1, which they show in this paper to regulate expression of IVR. In previous work they showed that SHE1 interacts with the 1a protein of cucumber mosaic virus (CMV). This was significant since transgenic expression of the 1a protein in tobacco plants inhibits viruses resistance. In this work they show that TMV infection of Samsun NN tobacco (which results in a hypersensitive response, HR) triggers increased SHE transcription accumulation. PVY, CMV and PVX caused only modest increases in the SHE1 transcript signal. Presumably because these viruses do not trigger the string HR resistance mechanism in Samsun NN. Engineered increased or decreased expression of SHE1 alters the size, number and morphology of TMV induced HR lesions on Samsun NN which increased expression enhancing resistance and decreased expression diminishing resistance to the virus. Examining TMV-GFP systemic movement in Samsun NN with increased/decreased SHE1 expression above the critical temperature range for N gene mediated resistance confirmed the role of SHE1 in resistance. At lower temperatures, decreasing the SHE1 level enhances virus spread. They confirm using Y2H, Co-IP and cell biology that SHE1 and IVR interact, suggesting an interaction with regulatory importance, and identify putative functional domains in the SHE1 protein primary sequence. The work is well carried out , the paper is well written and the findings are important for our understanding of plant virus resistance.
Minor corrections
Title : ... (IVR) in N Gene....
Corrected. Thank you.
line 273 temperature indicated?
Response: The temperature of the experiments was stated in the figure and Figure legend as being 33C.
